# Disability and Factors Associated with Disability in the Discharge Transition Phase After Acquired Brain Injury: An Observational Follow-Up Study

**DOI:** 10.3390/healthcare13161989

**Published:** 2025-08-14

**Authors:** Helene Honoré, Peter Preben Eggertsen, Asger Roer Pedersen, Jørgen Feldbæk Nielsen

**Affiliations:** 1Hammel Neurorehabilitation Centre & University Research Clinic (HNRC), 8450 Hammel, Denmark; peter.eggertsen@rm.dk (P.P.E.); joerniel@rm.dk (J.F.N.); 2Department of Clinical Medicine, Aarhus University, 8000 Aarhus, Denmark

**Keywords:** disability, quality of life, rehabilitation, occupational balance, return to work

## Abstract

**Background/Objectives:** Patients with acquired brain injury (ABI) often face complex challenges during the transition from in-hospital rehabilitation to everyday life. This study aimed to investigate disability, health-related quality of life (HRQoL), work, and other aspects of functioning as indicators of a meaningful life in this transition phase. Additionally, we assessed how disability three months post-discharge correlates with known risk factors. **Methods:** We conducted a prospective observational follow-up study including patients aged ≥18 years with ABI discharged from a specialized rehabilitation clinic. Patient-reported outcomes, including disability and HRQoL, were collected at discharge and three months later. Associations between disability and known risk factors were analyzed using multiple linear regression. **Results:** A total of 137 patients were included (mean age 63), with a follow-up completion rate of 59%. At follow-up, 11% reported complete recovery, while a moderate level of disability persisted overall, with no systematic change from discharge. HRQoL improved significantly, reaching a mean score of 0.83. Fatigue, sex, and time from injury to rehabilitation were significantly associated with disability levels. **Conclusions:** The transition phase after rehabilitation posed challenges for patients with ABI, with 38% experiencing moderate disability. Despite this, HRQoL improved to levels comparable with the general population. Fatigue, sex, and rehabilitation timing emerged as key factors influencing disability outcomes.

## 1. Introduction

Acquired brain injury (ABI) is defined as a collective term for non-congenital and non-degenerative brain lesions caused by traumatic or non-traumatic events [1], and it is a leading cause of disability among adults worldwide [2]. Patients with ABI present with physical, cognitive, and psychosocial challenges that may affect all aspects of life [3]. As an acute event, ABI may disrupt all parts of the patient’s life trajectory. ABI leads to extensive personal impact, such as reduced health-related quality of life and reduced satisfaction with life in general, for up to 61% [4]. This poses a potential challenge for patients who may strive to sustain a meaningful life. Kılınç and collaborators (2022) found that patients view energy as a form of treasured currency, which they manage carefully to participate in everyday life activities or other meaningful events. This balance is fundamental to living their lives and engaging in activities that enable them to achieve or maintain a sense of meaning and purpose [5].

### 1.1. Disability in the Discharge Transition Phase

Individual functioning reflects the patient’s lived experience and is widely regarded as the most important outcome of rehabilitation [6]. Conversely, disability represents the negative aspects of functioning and is equally essential to assess. More specifically, disability is the collective term for impairments, activity limitations, and participation restrictions. It reflects the negative aspects of the interaction between an individual’s health condition and contextual factors, including environmental and personal factors [7]. A review of 22 studies found that 24–49% of patients with ABI experience some level of disability, representing the most robust estimate to date [8]. To assess disability after ABI, it is essential to use a tool that captures the complexity of functioning across all life domains. The World Health Organization Disability Assessment Schedule 2.0 (WHODAS 2.0) [9] is increasingly recognized as the most robust tool measuring disability. It is grounded in the International Classification of Functioning, Disability, and Health (ICF) framework [7] and has been validated in neurological rehabilitation contexts [10,11,12]. It has demonstrated strong psychometric properties, including validity, reliability, and feasibility in both general and specific clinical populations [13,14,15]. WHODAS 2.0 has shown diagnostic accuracy in identifying moderate to severe disability after stroke [16]. WHODAS 2.0 has demonstrated cross-condition comparability in neurological populations [17], and its responsiveness to functional change supports its relevance in transitional rehabilitation settings [18]. These qualities make WHODAS 2.0 particularly suitable for capturing patient-reported disability in the critical transition from hospital to home, where functional challenges may emerge or evolve in everyday life contexts. In a Danish health-care perspective, the rehabilitation process is specifically aimed toward a meaningful life for the patient with the best possible extent of activity, participation, coping, and quality of life [19]. This calls for a salutogenetic approach in line with the WHO’s understanding of health as a state of complete physical, mental, and social well-being and not merely the absence of disease or infirmity [20]. Within this understanding, a meaningful life is not solely defined by functional independence but by the individual’s subjective experience of purpose, engagement, and participation in valued life roles. A recent scoping review further highlights that for adults with ABI, a meaningful life is characterized by being part of meaningful relationships, engaging in meaningful activities, and maintaining a coherent sense of identity despite changes in function and autonomy [21]. These findings underscore the importance of integrating patient perspectives into rehabilitation research and planning to support not only a functional recovery but also the reconstruction of meaning in everyday life. Patients with moderate and severe ABI typically begin the rehabilitation process in specialized in-hospital clinics [22]. They are later discharged to familiar social and physical surroundings, where they continue adjusting to their disability [23]. During this transition, they establish a new balance in their everyday life activities and discover to what degree they may recover, work, and live independently or with assistance. The extent of disability and dependency may not be clear to patients [24] or their relatives [25] at the time of discharge. Some challenges may only become apparent months later. Thus, the discharge transition phase implies a potentially stressful adjustment to disability during the change in environment [26]. During the discharge transition phase, patients have reported difficulties reengaging in meaningful activities [27].

### 1.2. Factors Affecting Disability in the Transition Phase

Previous research has identified several factors significantly associated with disability or related rehabilitation outcomes after ABI and thus as potential confounders of rehabilitation outcomes after ABI. These include age, sex, diagnosis, and the time between injury and referral to rehabilitation [28,29,30,31]. Fatigue has been linked to poorer short-term functional outcomes [32], a reduced likelihood of returning to work [33], and lower health-related quality of life [34]. A systematic review found that high self-efficacy is associated with better functioning in activities among patients with stroke [35]. This relationship is explained by the role of self-efficacy in enhancing personal effort, motivation, and persistence when facing barriers and setbacks [36]. In individuals aged 15–30 with ABI, both fatigue and functional independence were found to be significant predictors of disability [37]. Physical activity is associated with positive rehabilitation outcomes [38]. Also, cognitive dysfunction was found to be associated with low health-related quality of life after TBI [39].

This study aimed to investigate disability level and other indicators of a meaningful life for patients with ABI during the discharge transition from in-hospital rehabilitation. We also explored disability and factors associated with disability three months after discharge. Secondary outcomes included health-related quality of life, recovery, assistance requirements, hospital readmissions, work status, and occupational balance.

## 2. Materials and Methods

We set up a prospective follow-up study during the discharge transition phase. Baseline assessment was an interviewer-based survey one week prior to discharge, and follow-up was a self-reported survey three months after discharge.

Patients were recruited consecutively from Hammel Neurorehabilitation Centre and University Research Clinic, Denmark (HNRC) between September 2020 and September 2021, excluding a two-month period in early 2021 due to national COVID-19 restrictions. The HNRC is located in three cities: locations I-III (Figure 1). HNRC is a specialized neurorehabilitation facility operating across three locations in Central Denmark. It comprises ten inpatient units. All units follow a nationally standardized, evidence-based rehabilitation model. Rehabilitation is delivered by interdisciplinary teams consisting of physicians, nurses, physiotherapists, occupational therapists, and other specialists. The program is structured around four core principles: (1) a patient-centered approach, (2) a biopsychosocial treatment model, (3) interdisciplinary collaboration, and (4) a stepwise process involving comprehensive needs assessment, goal setting, intervention, and evaluation. This integrated model ensures consistency in care delivery while allowing for individualized treatment planning across all sites.

### 2.1. Participants and Recruitment

Patients admitted for more than two weeks were screened one week before discharge. The inclusion criteria included being 18 years or older, Danish-speaking, capable of giving informed consent, and able to complete a one-hour interview. Patients with low consciousness, severe aphasia, or other severe cognitive dysfunction were excluded. Patients showing symptoms such as emotional lability, agitation, or confusion were excluded for ethical reasons, as these conditions could be worsened by the interview process. Comorbidities like chronic obstructive pulmonary disease, diabetes, or other diseases did not lead to exclusion. Similarly, minor cognitive impairments, including attention deficits or mild aphasia, were not considered exclusion criteria. Cognitive function was assessed using the Functional Independence Measure (FIM) cognitive subscale. Patients with mild cognitive impairment were included, as these symptoms are frequently common after ABI. The study was designed to minimize disruption to patients’ ongoing rehabilitation activities. At each of the eight clinics, a designated staff member, such as a nurse, assistant, therapist, or secretary, evaluated patient eligibility based on inclusion criteria. The first author (HH) presented the study to eligible patients and invited them to participate voluntarily. If the patients consented, the baseline interview was conducted. Patients received a questionnaire link via email for the follow-up survey, with two reminder emails sent at one-week intervals. Given the potential vulnerability of patients post-ABI, no additional efforts were made to encourage participation. Patients received HH’s contact details in case they required technical assistance.

### 2.2. Baseline Characteristics

The first two authors extracted information on diagnosis, severity, comorbidity, and functional independence from discharge medical records (see Appendix A) by the first two authors. At baseline, HH assessed global disability using the Modified Rankin Scale (mRs), which ranges from 0 (no symptoms) to 6 (death) [40]. Patients self-reported anxiety and depression using the Hospital Anxiety and Depression Scale (HADS), which includes two subscales with seven items each, scored from 1 to 3, where 3 indicates the highest level. A subscale score above 8 (out of 21) indicates a moderate to high level of anxiety or depression [41].

### 2.3. Outcome Measures

Disability was the primary outcome of the study. Secondary outcomes were health-related quality of life, recovery, assistance requirements, hospital readmissions, work status, and occupational balance. A list of all the patient-reported outcome measures is provided in Appendix B.

#### 2.3.1. Disability

Disability was self-reported by patients via WHODAS 2.0 [9]. Each item was rated on a 5-point Likert scale from 0 (no difficulty) to 4 (extreme difficulty or inability to perform). At baseline, the 36-item version was administered by an interviewer, taking approximately 20 min. To reduce response burden, the 12-item self-administered version was used at follow-up. The 12-item version explains 81% of the variance in the 36-item version [9]. Disability scores from both assessments were standardized on a 0–100 scale, with 0 indicating no disability and 100 indicating complete disability [31], allowing for comparison of disability levels at baseline and follow-up. To aid interpretation, a categorization of WHODAS 2.0 has been suggested as follows: 0–4 (no difficulty), 5–24 (mild), 25–49 (moderate), 50–95 (severe), and 96–100 (extreme) [42].

#### 2.3.2. Covariates of Disability

Covariates included age, sex, ABI type, time from injury to rehabilitation, self-efficacy, premorbid physical activity level, pathological fatigue, and cognitive functional independence. The General Self-Efficacy Scale (GSES), a 10-item tool measuring optimistic beliefs in handling life challenges, was used to assess self-efficacy [43]. Scores ranged between 10 and 40, with higher values indicating greater self-efficacy. Physical activity level was categorized using the Saltin-Grimsby Physical Activity Level Scale [44]. Fatigue was self-reported with the Multidimensional Fatigue Inventory (MFI) [45], which includes five subscales scored from 4 (no fatigue) to 20 (severe fatigue). A general fatigue subscore of 12 or higher was considered indicative of pathological fatigue [46]. Cognitive function was rated using the FIM subscale for cognition, with scores from 1 (total assistance) to 7 (complete independence), yielding a total between 5 and 35 [47].

#### 2.3.3. Secondary Outcome Measures

Health-related quality of life was measured using the EQ5D-5L, calculated with a Danish value set ranging from −0.757 (worse than death) to 1 (full health) [48]. The EQ5D-5 L describes health-related quality of life across five dimensions anxiety/depression, pain/discomfort, mobility, self-care, and usual activities on five levels (no problems, slight problems, moderate problems, severe problems, and unable/extreme problems).

Work status was reported in the following categories: self-employed, employed, unemployed, retired, studying, and on sick leave. These were dichotomized by grouping the responses “not employed”, “retired”, “studying”, and “on sick leave” into a single category labeled “not in remunerative work”.

Occupational balance was assessed with the Danish Occupational Balance Questionnaire, OBQ-DK [49,50]. We used the 11-item version, with four ordered responses ranging from “disagree” = 0 and “partially disagree” = 1 to “partially agree” = 2 and “agree” = 3. The total sum score ranges from 0 to 33, with higher scores indicating better occupational balance. The OBQ-DK summary score was dichotomized into low and high occupational balance using a cut-off at <22, indicating that the patient could have “agreed” (score 3) with at least one item. Thus, the cut off value was calculated as more than the total number of items (11) × score 2.

Recovery and personal assistance requirements were assessed using the “Two simple questions” [51,52]. The extent of community-based rehabilitation and hospital re-admissions was assessed using customized questions.

### 2.4. Data Analysis and Statistics

Patient characteristics were presented as frequencies (n) and percentages (%). Functioning scores were reported within the summary range of individual scores. Demographic and health parameters of included patients were compared with those of excluded patients at the HNRC.

Baseline disability was reported using both the WHODAS 2.0 median scores for each domain and the summary median score. The summary score was calculated as the weighted complex scoring method presented in the manual and reported as an index score from 0–100 [9]. Follow-up disability was also reported as an index score of 0–100. It was calculated by dividing the summary score by the maximum score (48) and multiplying by 100, in accordance with the manual [9]. For cases with fewer than two missing WHODAS 2.0 items, imputation was performed using the median score [9]. Multiple linear regression analyses were conducted with follow-up disability as the outcome. Model assumptions were evaluated using plots of observed versus predicted values, scatter plots, residual plots, histograms, and QQ plots. Statistical analyses were performed in Stata 19.0 (StataCorp, College Station, TX, USA).

Secondary outcomes were summarized descriptively and visualized using plots of median scores. Change in work status from baseline to follow-up was tested with McNemar’s test. The Wilcoxon rank-sum test was used to compare disability levels for patients in remunerative work and those not, as well as between patients reporting potential occupational balance versus imbalance.

## 3. Results

### 3.1. Patient Characteristics

Of 586 screened patients, 137 were included, 62% males, and the mean age was 63 years. The cohort is presented in Table 1. Patients began rehabilitation a median of 10 days following their ABI. The largest group represented was patients with stroke (77%). Patients included in the study differed significantly in cognitive function, as measured by the FIM subscale (*p* < 0.01), compared to those not included.

### 3.2. Disability in the Discharge Transition Phase for the Respondents

The survey answers at baseline are presented in Table 2. See Appendix C for visualized data plots of variables from Table 2 for visual inspection. At discharge, the median disability level was 34%, with ‘life activities’ being the most impacted domain (median 60%).

Raw estimates of the patients’ survey answers at follow-up are presented in Table 3. Median disability at follow-up was 30% (WHODAS 2.0), showing no significant change from discharge (*p* = 0.4860). At follow-up, 59% of patients had disability scores indicating moderate to severe impairment (raw data plotted in Figure 2).

The covariates of disability are plotted in Figure 3. Sex, time from ABI to rehabilitation, and pathological fatigue were significantly associated with disability at follow-up (see Appendix D for adjusted estimates). After adjustment, patients who began rehabilitation 28–44 days post-ABI had disability scores 20.61 points higher than those who started within 1–7 days (*p* = 0.035). Patients with pathological fatigue had disability scores 9.60 points higher than those without fatigue (*p* = 0.023). Men had significantly lower disability scores than women, with a difference of −8.39 points (*p* = 0.049).

### 3.3. Other Aspects of Functioning in the Discharge Transition Phase

Complete recovery was reported by 11% of participants. At follow-up, 62% of patients lived independently, and 71% were engaged in community-based rehabilitation programs with physiotherapy, occupational therapy, speech therapy, or group-based training. Health-related quality of life improved significantly from discharge, reaching 0.83 (SD 0.21) at follow-up (mean change 0.17; 95% CI 0.12–0.23; *p* < 0.001). The unadjusted estimates of disability and health-related quality of life are plotted in Figure 2.

#### 3.3.1. Work Status

Employment rates declined significantly from 41% pre-ABI to 13% at follow-up (*p* < 0.001; see Table 3). Of the 45 respondents below retirement age (65 years) in the cohort, 18% were working and reported mild disability levels at follow-up (Figure 4), whereas 10% reported being retired, 16% were unemployed, and 44% were on sick leave. Non-working patients had significantly higher disability scores compared to those in paid employment (*p* = 0.0030).

#### 3.3.2. Occupational Balance

At follow-up, 52% of patients reported occupational balance, with a median score of 23 (IQR 9). Potential imbalance was reported by 40% of patients on at least one item. For patients with responses classified as potential imbalance, the disability level ranged wider but it did not differ statistically from patients with responses classified as potential balance (*p* = 0.013) (see box plot in Figure 5).

### 3.4. Patients at Follow-Up

At follow-up, the response rate ranged from 41 to 47% (Table 3). An overview of the patients’ demographics, diagnoses, and functioning at baseline was provided for patients who remained in the cohort at follow-up (right column) and who dropped out (middle column) for easy comparison (Table 2). Respondents at follow-up were younger than non-respondents with a mean age of 60 versus 68 years, and they reported more frequently cohabiting with another adult at discharge than non-respondents. Respondents had higher functional levels measured with FIM scores and a WHODAS 2.0 disability index than those lost to follow-up. The summary score was higher for the non-respondents with a median of 35 vs. 33 for participants. All domain scores for disability were higher for non-respondents except for the domains “getting along” and “social participation”. The largest difference in disability was in the cognition domain with a median of 5 for respondents and 15 for non-respondents. FIM scores on cognition were also lower for respondents than for non-respondents. Corresponding estimates were found for respondents and non-respondents regarding baseline level for sex, depression, fatigue, and general self-efficacy scores. Also, the distribution of ABI diagnoses was similar between the two groups, with a prevalence of apoplexy of 77–79%.

## 4. Discussion

We evaluated disability and related aspects of functioning in 137 patients during the discharge transition phase following ABI. Follow-up data were obtained from 59% of the original cohort. Complete recovery was reported by 11% after three months. Moderate disability levels remained stable from discharge to follow-up, with no significant change observed. A median disability level of 38% was found when adjusted for confounding variables. Health-related quality of life improved from 0.63 (0.35) at discharge to 0.83 (0.21) at follow-up with a mean improvement of 0.17 (95% CI 0.12–0.23), *p* < 0.001. During the transition phase, 13% of patients resumed employment, and they reported lower disability levels than non-working patients (*p* = 0.0030). Twenty-nine patients (40%) reported potential imbalance.

### 4.1. Comparison with Other Studies

Disability levels in this cohort were comparable to the 28–44% range reported in patients with stroke at the time of discharge [8]. Longer-term studies (≥9 months) have consistently shown a reduction in disability over time [8], indicating that functioning improves over time after sub-acute rehabilitation. The stationary disability level in our study may reflect the relatively short follow-up period. Also, disability is context-dependent, influenced by the patients’ surroundings, and as the patients might take on more complex and challenging tasks at home than in hospital settings, the transition phase could lead to unchanged disability levels for patients even with increased functional levels. With 71% still participating in community-dwelling rehabilitation, a potential for reduction in disability could be aspired.

Confirmed confounders of disability in our study were sex, fatigue, and time to rehabilitation admission. Male patients reported lower disability scores than female patients. This aligns with previous findings showing higher disability rates among women, along with higher prevalence of depression and dependency in everyday life across several populations [53,54]. Time between ABI onset and rehabilitation showed the strongest confounding effect. Acute care duration may serve as an indirect indicator of ABI severity or complexity, which would match the higher level of disability.

We were surprised to find no confounding effect of self-efficacy. The level of self-efficacy reported in our study (median = 33; IQR = 9) was higher than the self-efficacy level reported in a group of community-dwelling people with stroke after a neuropsychological rehabilitation intervention (mean = 30, sd  =  7) [55]. This suggests that patients felt confident in managing new challenges and adversity. The level of perceived self-efficacy has been identified in a qualitative study to be the foundation for a successful transition experience [56] and a systematic review linked high self-efficacy to higher levels of functioning in daily activities in stroke patients [35]. However, this was not confirmed in our study. One explanation may be that the patients in rehabilitation may be in a hopeful state of mind due to ongoing progress.

#### 4.1.1. Health-Related Quality of Life

The health-related quality of life at follow-up was 0.83, aligning closely with the Danish population norm of 0.82 [49]. The observed statistically significant change in health-related quality of life of 0.12 (*p* < 0.001) was at a clinically important level [57,58]. This finding is notable given that disability levels remained unchanged over the same timeframe. By contrast, disability level was identified as a primary factor in reducing quality of life among stroke patients [59]. Health-related quality of life scores in stroke populations vary widely (from −0.02 to 0.92) and do not correlate linearly with illness severity [60]. Barclay-Goddard and collaborators (2012) proposed that health-related quality of life may improve over time as individuals adapt and re-normalize to long-term disability [61,62]. In a rehabilitation perspective, the high quality of life found in the cohort is encouraging. It has been suggested that enhancing quality of life rather than reducing impairment should be the primary goal of rehabilitation because it contributes more directly to the individual experience of meaning in everyday life [63,64]. Recent research has shown strong correlations between WHODAS 2.0 and EQ5D-5L scores in ICU survivors, suggesting that the tools assess overlapping constructs such as mobility, self-care, and life activities [65]. However, WHODAS 2.0 uniquely captures domains like cognition and participation, which are particularly relevant in ABI populations. These insights support the complementary use of WHODAS 2.0 in assessing broader aspects of functioning beyond health-related quality of life. In our study, this distinction may help explain why health-related quality of life improved significantly while disability levels remained unchanged. It underscores the importance of using multidimensional tools like WHODAS 2.0 to capture the full scope of patients’ lived experiences during the transition phase after ABI.

#### 4.1.2. Work

Three months post-discharge, just 11% had returned to remunerative employment. The most robust available estimation of return to work after ABI was calculated in a 2009 systematic review of 49 studies, which found that 40% return to work, and they described lower proportions of return to work with shorter time to follow-up [66]. This suggests that returning to work is generally challenging for people with ABI. A mixed-methods study identified stigma, adjustment, support, and readiness as key factors affecting return-to-work outcomes after ABI. The study recommended workplace strategies such as disability awareness and fatigue management plans [67]. Given that fatigue significantly influenced disability and that working patients had lower disability levels at follow-up, our results reinforce the importance of addressing these factors in vocational rehabilitation. Fatigue has consistently been identified as a significant predictor of disability in ABI populations. Worm et al. found that self-reported fatigue and cognitive difficulties were strongly associated with higher disability scores in adolescents and young adults post-ABI [37], underscoring the importance of assessing and managing these symptoms in ABI rehabilitation.

#### 4.1.3. Occupational Balance

Twenty-nine patients (40%) reported potential imbalance with a summary median score of 23 (IQR 9). Occupational balance was previously reported in a stroke population with patients 1–4 years after stroke [68], with over 50% of participants indicating poor occupational balance. Comparing results across studies is challenging due to the absence of standardized cut-off values for defining good or poor occupational balance. Additionally, earlier studies used a 13-item version of the OBQ, differing from the 11-item version applied here. When converted to an index score (OBQ sum score/max score × 100), the summary score of 23 in the present study is equal to an index of 70. This indicates a notably high level of occupational balance compared to all previous studies, which reported findings equal to index values from 29 to 51 [68,69,70]. Our findings are surprising because the moderate disability level might typically suggest difficulties in prioritizing meaningful activities. Recent research has raised concerns about the validity of the OBQ-DK in populations with ABI. These concerns suggest that the unmatched results observed in our study may, at least in part, be attributed to measurement limitations rather than actual differences in occupational balance [71]. Nonetheless, the high occupational balance matches the cohort’s elevated level of quality of life and may reflect strong coping skills regarding disability. The study was conducted during the COVID-19 pandemic, a period marked by heightened uncertainty and restrictions. These circumstances may have influenced patients’ emotional states and responses to self-reported measures such as HADS, WHODAS 2.0, and EQ5D-5L, potentially confounding the assessments but also potentially inhibiting at-home interventions by rehabilitation professionals.

### 4.2. Clinical Implications—A Meaningful Life in the Discharge Transition Phase

In a rehabilitation aimed at supporting a meaningful life for the patients—with the best possible extent of activity, participation, coping, and quality of life—it is concerning that disability levels remained moderate throughout the discharge transition phase. Notably, the domain “life activities” was the most affected with a median disability of 60%. On the other hand, the high level of health-related quality of life may reflect an acceptance of activity limitations and participation restrictions. The insight that health-related quality of life can improve significantly to a high level in spite of persisting disability is crucial for clinicians, as it helps sustain hope that the patients can experience a meaningful life even in the absence of substantial functional recovery after ABI.

This study highlights the need for continued, individualized rehabilitation support following discharge, particularly addressing cognitive and participatory domains of functioning. The association between fatigue and disability suggests that systematic fatigue screening and management should be integrated into post-ABI care. Gender differences in disability outcomes may warrant tailored rehabilitation strategies. Finally, the high levels of occupational balance and quality of life, despite persistent disability, underscore the importance of person-centered approaches that promote adaptation and engagement in meaningful activities.

This study contributes to the field by operationalizing indicators of meaningful life through the combination of outcome measures of disability, health-related quality of life, occupational balance, and return to work within the discharge transition phase. However, as highlighted in a recent scoping review [21], a meaningful life after ABI is not merely a function of restored capacity but is deeply rooted in the individual’s ability to engage in valued relationships and activities and to maintain a coherent sense of identity despite altered circumstances.

The observed dissociation between stable disability levels and improved quality of life in our cohort underscores the need to move beyond traditional physical outcome metrics and embrace a more person-centered, biopsychosocial understanding of rehabilitation success.

### 4.3. Limitations

This study has several limitations. First, the risk of selection bias must be acknowledged. Although the included cohort was largely comparable to excluded patients at HNRC, a statistically significant difference in cognitive function (FIM cognitive subscale) was observed. This was expected, given that the inclusion criteria required patients to participate in a one-hour interview. Cognitive function likely influenced the self-reported outcome measures on anxiety and depression, fatigue, disability, and health-related quality of life.

A major limitation of this study is the absence of a formal assessment of awareness or anosognosia. In individuals with ABI, impaired self-assessment of competence may contribute to under-reporting of symptoms and over-reporting of functioning, especially in the early rehabilitation stages [72]. Awareness can be ethically assessed using structured tools that respect patient readiness and autonomy, such as the Motor Unawareness Assessment (MUNA) [73]. Such assessments can be applied in an indirect and non-confrontational way as described by Berti [74], Bisiach [75], and collaborators. Including such a measure would have allowed for an interpretation of the potential impact of impaired self-awareness on self-reported disability, while still respecting the patient’s readiness, emotional safety, and autonomy [76]. Although we considered including a dedicated awareness measure, we found that no standardized awareness assessment tool was validated in a Danish context at the time of the study. This may have introduced bias into our findings. In hindsight, adding a proxy-reported disability assessment could have mitigated the limitation. Proxy assessments offer valuable complementary perspectives for the interpretation of findings in disability studies.

A study comparing disability scores from WHODAS 2.0 from both stroke patients and proxies found higher disability levels reported by proxies [77]. Similar findings were established in psychiatric settings [78] and in patients with cerebral palsy [79]. However, the patient and proxy measures correlated well, and the 12-item version of WHODAS 2.0 was recommended for discharge assessment of disability [77]. We therefore recommend that future studies on rehabilitation trajectories incorporate parallel proxy measures to enhance the validity and interpretability of self-reported outcomes.

The drop-out rate of 41% was consistent with similar follow-up studies [80,81]. Respondents at follow-up reported lower disability and higher health-related quality of life at discharge compared to non-respondents. This may have resulted in under-estimation of disability and over-estimation of health-related quality of life.

Finally, only patients with moderate or severe ABI who received rehabilitation at HNRC were included, limiting the generalizability of findings to patients with ABI of the same severity and from comparable healthcare systems.

## 5. Conclusions

The present study underscores the complexity of the discharge transition phase for patients with ABI, where persistent moderate disability may challenge the pursuit of a meaningful life. Encouragingly, health-related quality of life improved to levels comparable to the general Danish population, offering hope to patients and rehabilitation professionals alike. Fatigue, sex, and time from injury to rehabilitation were confirmed as significant factors influencing disability three months post-discharge, with consideration for potential selection bias. Findings are easily generalized to other Danish or similar rehabilitation contexts in Scandinavia, and further international studies are warranted to examine how rehabilitation during the transition phase influences recovery trajectories for ABI patients across diverse healthcare systems.

## Figures and Tables

**Figure 1 healthcare-13-01989-f001:**
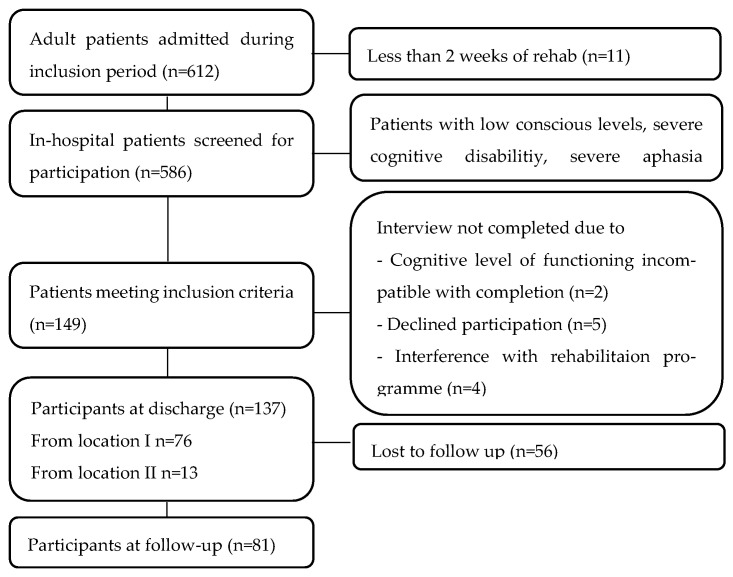
Flowchart showing recruitment, participation, and drop-out.

**Figure 2 healthcare-13-01989-f002:**
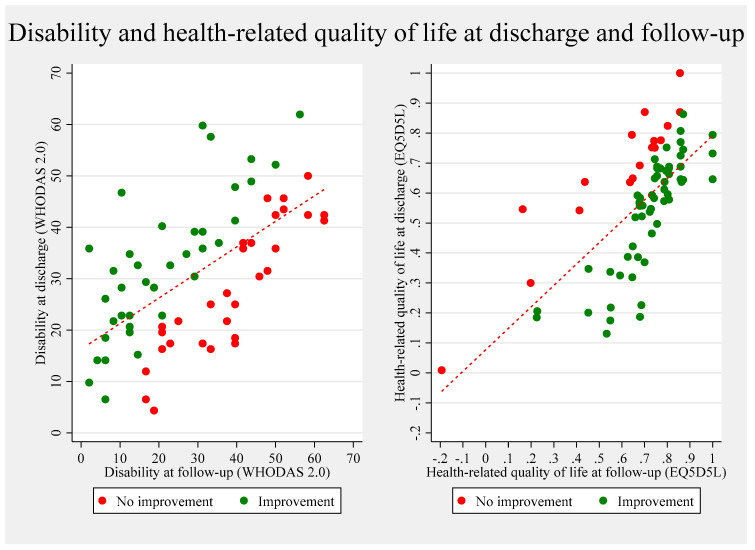
Unadjusted (raw) scores of disability and health-related quality of life at discharge plotted against scores at follow-up. The two plots illustrate individual-level changes in disability (WHODAS 2.0, **left** panel) and health-related quality of life (EQ-5D-5L, **right** panel) from discharge to three-month follow-up. Each point represents a participant. Green dots indicate improvement, while red dots denote no improvement or deterioration. The diagonal line represents equality between discharge and follow-up scores, and the fitted red line shows the overall trend. All estimates are unadjusted.

**Figure 3 healthcare-13-01989-f003:**
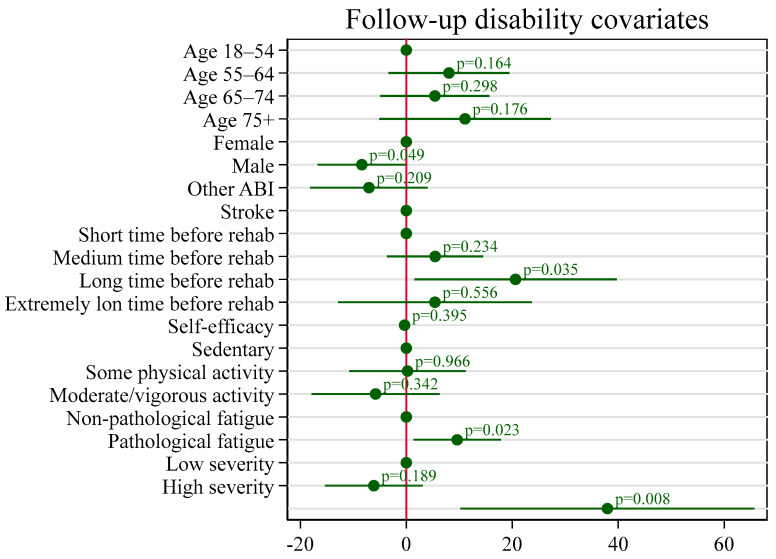
Coefficient plot displaying the associations between covariates and follow-up disability. The plot presents estimated coefficients from the regression model, with 95% confidence intervals. Covariates include age group, sex, type of acquired brain injury (ABI), time from injury to rehabilitation admission, self-efficacy (GSES), pre-injury physical activity level, fatigue status, and cognitive functioning (Functional Independence Measure Cognitive subscore). The reference categories are indicated by the base levels, and *p*-values are annotated next to each coefficient. A vertical line at zero indicates the null effect.

**Figure 4 healthcare-13-01989-f004:**
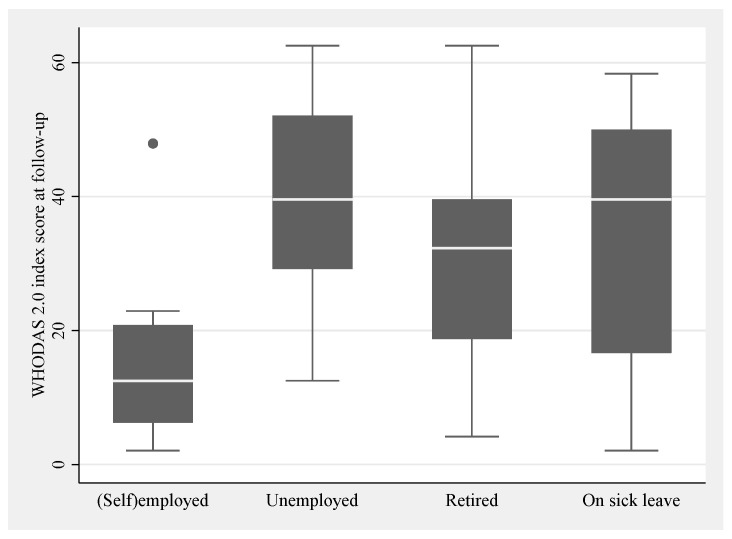
Work status and disability level at follow-up. Box plots with whiskers show the categorical classification of work status as self-employed or employed (=working) and not working (unemployed, retired or on sick leave) at follow-up on the horizontal axis and numerical scores on the vertical axis of disability at follow-up. The boxes indicate upper and lower quartiles with the median marked as a line inside. One dot is shown as an outlier.

**Figure 5 healthcare-13-01989-f005:**
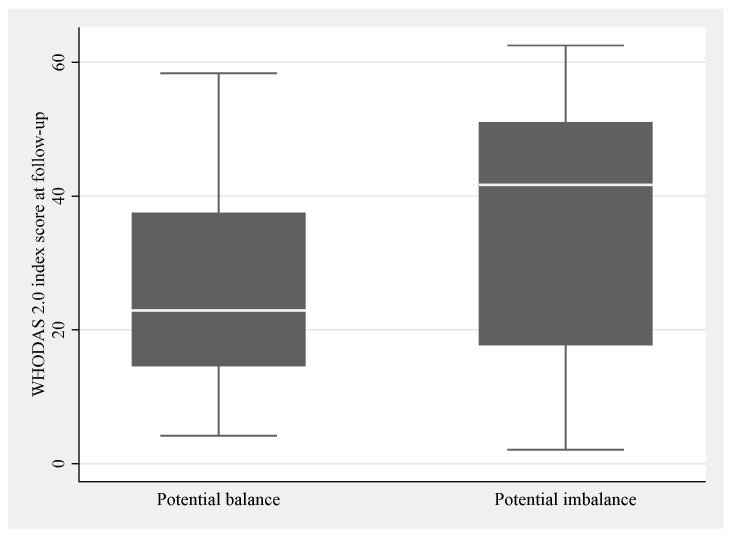
Occupational balance and disability level at follow-up. Box plots showing the categorical classification of occupational balance on the horizontal axis as potential balance and imbalance, and numerical scores on the vertical axis of disability at follow-up. The boxes indicate upper and lower quartiles with the median marked as a line inside.

**Table 1 healthcare-13-01989-t001:** Demographics, diagnosis, and severity at onset for included and excluded patients.

Characteristics		Included n = 137	Excluded n = 449	*p*-Value
Age in years ^a^		63 (13)	62 (14)	0.72 ^1^
Sex male ^c^		86 (62)	268 (60)	0.69 ^2^
Diagnosis and severity at onset				
ABI diagnosis ^c^	Stroke	106 (77)	334 (74)	0.98 ^2^
	SAH	6 (4)	26 (6)	
	Trauma	8 (6)	27 (6)	
	Encephalopathy	6 (4)	23 (5)	
	Other	11 (8)	39 (9)	
Charlson Comorbidity Index ^b^		5.13 (4.12)	5.97 (6.52)	0.56 ^2^
Days from ABI to rehab admission ^b^		10 (15)	13 (18)	0.85 ^3^
Rehab days ^b^		51 (29)	51 (30)	0.97 ^3^
Functional independence measure ^b^	Cognitive	27.5 (20)	26 (22)	<0.01 ^3^
	Motor	81 (59)	81 (78)	0.87 ^3^

Continuous data presented with ^a^ mean (standard deviation) or ^b^ median (inter quartile range), and categorical data presented with ^c^ n (%). ABI = acquired brain injury; SAH = subarachnoid haemorrhage. Functional Independence Measure Cognitive subscore range 5–35, Motor subscore range 13–91. ^1^ *t*-test, ^2^ Fishers Exact test, ^3^ Wilcoxon Rank-sum test.

**Table 2 healthcare-13-01989-t002:** Descriptive patient characteristics at baseline at discharge.

			Baseline (n = 137)	Participants at Follow-Up (n = 81)	Lost to Follow-Up (n = 56)
Pre-morbid characteristics		Possible range			
Age in years ^a^		-	63 (13)	60 (13)	68 (13)
Sex male ^c^		Yes/No	86 (62)	50 (62)	35 (63)
Unassisted living ^c^		Yes/No	131 (96)	78 (96)	53 (95)
Cohabiting ^c^		Yes/No	98 (72)	61 (75)	37 (66)
Physical activity level (SG) ^c^	Sedentary	Yes/No	37 (27)	19 (23)	18 (32)
	Some physical activity	Yes/No	59 (43)	37 (46)	22 (39)
	Moderate physical activity	Yes/No	39 (29)	23 (28)	16 (29)
	Vigorous physical activity	Yes/No	2 (1)	2 (2)	0 (0)
Work status ^c^	(Self-)employed	Yes/No	56 (41)	38 (47)	18 (32)
	Unemployed	Yes/No	5 (4)	4 (5)	1 (2)
	Retired	Yes/No	68 (50)	32 (40)	36 (64)
	Student	Yes/No	1 (1)	1 (1)	0 (0)
	On sick leave	Yes/No	7 (5)	6 (7)	1 (0)
Diagnosis, severity and duration					
ABI diagnosis ^c^	Stroke	Yes/No	106 (77)	62 (77)	44 (79)
	SAH	Yes/No	6 (4)	5 (6)	1 (2)
	Trauma	Yes/No	8 (6)	6 (7)	2 (4)
	Encephalopathy	Yes/No	6 (4)	2 (2)	4 (7)
	Other	Yes/No	11 (8)	6 (7)	5 (9)
ABI severity ^c^	Mild	Yes/No	69 (50)	43 (53)	26 (47)
	Moderate	Yes/No	35 (26)	21 (26)	14 (25)
	Severe	Yes/No	33 (24)	17 (21)	16 (29)
Days from ABI to rehab admission ^b^		-	10 (15)	10 (13)	10 (22)
Rehab days ^b^		-	43 (34)	42 (44)	46 (28)
Health and functioning					
Charlson Comorbidity Index ^b^		0–100	3 (3)	3 (3)	3 (3)
Functional independence measure ^b^	Cognitive	5–13	28 (7)	29 (7)	26 (9)
	Motor	13–91	81 (15)	83 (13)	80 (17)
Modified Rankin Scale score (mRs) ^c^	No symptoms	Yes/No	2 (1)	1 (1)	1 (2)
	No significant disability	Yes/No	44 (32)	24 (30)	20 (36)
	Slight disability	Yes/No	35 (26)	27 (33)	8 (14)
	Moderate disability	Yes/No	29 (21)	14 (17)	15 (27)
	Moderate-severe disability	Yes/No	26 (19)	14 (17)	12 (21)
	Severe disability	Yes/No	1 (1)	1 (1)	0 (0)
Walking ability ^c^	Able to walk 10 m	Yes/No	107 (78)	65 (80)	42 (75)
Anxiety (HADS)	Sum score ^b^	0–21	3 (5)	3 (5)	3 (4)
	Moderate-high level ^c^	Yes/No	23 (16)	15 (18)	8 (14)
Depression (HADS)	Sum score ^b^	0–21	4 (4)	4 (4)	4 (4)
	Moderate-high level ^c^	Yes/No	22 (16)	14 (18)	8 (14)
Fatigue (MFI)	General fatigue ^b^	4–20	12 (8)	12 (9)	12 (9)
	Physical fatigue ^b^	4–20	12 (8)	12 (9)	12 (7)
	Reduced activity ^b^	4–20	12 (9)	12 (9)	13 (7)
	Reduced motivation ^b^	4–20	7 (5)	7 (5)	6 (5)
	Mental fatigue ^b^	4–20	8 (9)	8 (7)	8 (11)
	Pathological fatigue ^c^	Yes/No	72 (53)	44 (54)	28 (50)
General self-efficacy (GSES) ^b^		0–40	33 (9)	33 (8)	33 (10)
Health-related quality of life (EQ5D-5L) ^b^		−0.757–1	0.63 (0.35)	0.67 (0.32)	0.55 (0.34)
Disability (36-item WHODAS 2.0) ^b^	Cognition	0–100	10 (30)	5 (25)	15 (40)
	Mobility	0–100	31 (56)	31 (56)	38 (56)
	Self-care	0–100	20 (40)	20 (50)	30 (50)
	Getting along	0–100	17 (25)	17 (25)	17 (25)
	Life activities	0–100	60 (60)	60 (60)	65 (60)
	Social participation	0–100	38 (25)	38 (21)	38 (21)
	Summary score	0–100	34 (25)	33 (22)	35 (27)

Data presented for all included patients, for respondents and non-respondents at follow-up. Mean (standard deviation) ^a^; median (inter quartile range) ^b^, n (%) ^c^. The category self(employed) covered all patients with labor attachment including seven on sick leave. SG = Saltin-Grimsby Physical Activity Level Scale; ABI = acquired brain injury; GCS = General Self-efficacy scale; HADS = Hospital Anxiety and Depression Scale; MFI = Multidimensional Fatigue Inventory; GSES = General Self-efficacy Scale; EQ5D-5L = EuroQuol quality of life index; WHODAS 2.0 = World Health Organization Disability Assessment Schedule 2.0.

**Table 3 healthcare-13-01989-t003:** Descriptive data on self-reported functioning at follow-up.

Self-Reported Functioning at Follow-Up, n = 81		Possible Range	Estimate	Missing
Complete recovery ^c^		Yes/No	9 (11)	0
Any assistance required ^c^		Yes/No	31 (38)	0
Community-based rehab ^c^		Yes/No	55 (71)	4
Hospital readmission ^c^		Yes/No	8 (11)	6
Work status ^c^	(Self-)employed	Yes/No	10 (13)	5
	Unemployed	Yes/No	8 (11)	5
	Retired	Yes/No	38 (50)	5
	On sick leave	Yes/No	20 (26)	5
Occupational balance	OBQ-DK sum ^b^	0–33	23 (9)	9
	Potential good balance ^c^		43 (60)	9
Health-related quality of life (EQ5D-5L) ^b^		−0.757–1	0.83 (0.21)	6
Disability (12-item WHODAS 2.0) ^b^		0–100	30 (25)	9

Median (inter quartile range) ^b^ and proportions n (%) ^c^. OBQ-DK = Occupational Balance Questionnaire Danish 11-item version summary score (range 0–33) and proportion of potential good balance with a OBQ-DK sum score ≥ 23. EQ5D-5L = EuroQuol Five-Dimensional Health-related Quality of Life Assessment Tool, WHODAS 2.0 = World Health Disability Assessment Schedule 2.0.

## Data Availability

The data presented in this study are not publicly available due to ethical and legal restrictions related to the protection of patient privacy and confidentiality.

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
