# Peer review of "Disability and Factors Associated with Disability in the Discharge Transition Phase After Acquired Brain Injury: An Observational Follow-Up Study"

_healthcare, 2025, doi:10.3390/healthcare13161989_

Round 1

Reviewer 1 Report

Comments and Suggestions for Authors

In this paper, the authors investigated the disability and health-related quality of life of 137 patients with ABI at the hospital discharge and three months post-discharge follow-up.

The overall quality of the paper is good in terms of paper structure, English grammar and methodological approach. A strength point of the paper is the high numerosity of the enrolled patients (137) and experimenters’ effort in monitoring patients’ status along all the period.

The unique weak point is represented by data visualization that could be easily improved with few modifications.

Here are my comments:

-I suppose there was a problem on the first page of the manuscript since part of the Introduction is in italic;

-I suggest the authors reduce the number of tables and replace them with graphical representation in order to improve manuscript readability and data visualization. An instance could be substituting Table 2 with a boxplot;

-Since the authors assessed the patients’ disability and quality of life at the discharge and at three months post discharge, I suggest reporting some plots demonstrating the variations of the same indicator (i.e., WHODAS and EQ5D-5L score between these two-time frames).

Author Response

General Response to All Reviewers

We sincerely thank all three reviewers for their detailed and constructive feedback. Your comments were not only insightful but also phrased in a collegial and encouraging tone. We found your suggestions to be highly relevant and well-informed, and they provided a solid foundation for improving the manuscript. We have carefully addressed each point and revised the manuscript accordingly. We hope the changes reflect your valuable input and enhance the overall quality and clarity of the study.

Response to Reviewer 1

Comment: Part of the Introduction is in italic.

Response: Thank you for pointing this out. The formatting error has been corrected.

Comment: Consider replacing Table 2 with graphical representations to improve readability.

Response: We appreciate this excellent suggestion. While we retained Table 2 for comparability with similar studies, we have added graphical representations of all variables in Appendix C to enhance data visualization and transparency, and we decided to visualize the results from former Table 3 in a Figure showing coefficients.

Comment: Suggest adding plots to show changes in WHODAS and EQ5D-5L scores over time.

Response: Thank you for this constructive idea. We have added scatterplots in Figure 2 to illustrate individual-level changes in WHODAS and EQ5D-5L scores from discharge to follow-up.

Reviewer 2 Report

Comments and Suggestions for Authors

Dear Editors,

Thank you for inviting me to act as a Reviewer for the Journal of Healthcare regarding the manuscript entitled 'Disability and factors associated with disability in the discharge transition phase after acquired brain injury. An observational follow-up study'. The study is interesting and aims to investigate disability, health-related quality of life (HRQoL), work, and other aspects of functioning as indicators of a meaningful life in this transition phase. Additionally, the study assessed how disability levels three months post-discharge correlate with known risk factors. This study delivers essential insights into the experience of patients with acquired brain injury (ABI) during the critical transition from inpatient rehabilitation to community living. It makes a meaningful contribution to the literature on post-ABI rehabilitation outcomes. Among the strengths of this study should be counted the use of validated tools (WHODAS 2.0, EQ5D-5L, GSES, MFI) and a cohort from a well-established rehabilitation center, which enhances the study's credibility, as well as the well-addressed disconnection between disability and HRQoL. Among the weaknesses of this study are the following: 

  My recommendation is major revisions and reconsideration after that stage. In particular:

  • In lines 30-41 and also in some other parts of the manuscript, the fonts used are different from the rest of the text. This is insignificant but should be corrected. Generally, there are minor grammatical errors and occasional tense inconsistencies. A full proofread is recommended before final submission.
  • The introduction section is limited and lacks flow. The introduction section should be slightly expanded based on recent references.
  • The term' acquired brain injury', which is the main term of the manuscript, should be defined and further explained.
  • The definition of disability in lines 42-44 should be primarily based on the definition provided by the World Health Organization.
  • The aims of the study should be more clearly presented. The proofreading could help in that direction.
  • A qualitative complement to the referred concepts, such as "meaningful life," could have deepened the analysis.   
  • In lines 129-134, the authors explain that they have used two versions of Whodas: the extended version, comprising 36 questions, and the short version, containing 12 questions, with the former used for the baseline assessment and the latter for the follow-up assessment. The question is, why haven't the authors used the short version in both phases, which would be methodologically correct?
  • In 2.3.2 and 2.3.3, the research tools are not sufficiently described. Instead, there is a very short, mixed, and confusing description. A clearer and more analytical description could enhance the readability of this article.
  • In the discussion section, the practical implications should be further described.
Yours sincerely, Comments on the Quality of English Language

Dear Authors,

I suggest the manuscript to be professionally proofread, to ensure cohesion and clarity. 

Author Response

General Response to All Reviewers

We sincerely thank all three reviewers for their detailed and constructive feedback. Your comments were not only insightful but also phrased in a collegial and encouraging tone. We found your suggestions to be highly relevant and well-informed, and they provided a solid foundation for improving the manuscript. We have carefully addressed each point and revised the manuscript accordingly. We hope the changes reflect your valuable input and enhance the overall quality and clarity of the study.

Response to Reviewer 2

Comment: Font inconsistencies and minor grammatical errors throughout.

Response: The manuscript has undergone thorough proofreading, correcting all formatting issues, grammar, and tense inconsistencies. We also rephrased complex sentences to improve clarity.

Comment: The introduction lacks flow and should be expanded.

Response: The Introduction has been expanded and restructured for better readability and flow. Recent references have been added to strengthen the background.

Comment: Define “acquired brain injury.”

Response: A clear definition of ABI has been added in the first paragraph of the Introduction.

Comment: Use WHO’s definition of disability.

Response: The definition of disability now aligns with the ICF framework from the World Health Organization.

Comment: Clarify study aims.

Response: The aims are now clearly stated at the end of the Introduction.

Comment: Consider a qualitative complement to concepts like “meaningful life.”

Response: We appreciate the reviewer’s suggestion to include a qualitative complement to concepts such as “meaningful life.” Due to space constraints and the need to prioritize content, we had to omit further elaboration on meaningfulness in this version of the manuscript. However, we have revised both the Introduction and Discussion sections to better frame the concept and hope the reviewer will find these revisions constructive.

Comment: Why use different WHODAS versions at baseline and follow-up?

Response: We appreciate the reviewer’s thoughtful observation regarding the use of two different WHODAS 2.0 versions. At baseline, we chose the 36-item version to gain a detailed understanding of which specific domains of functioning were most affected at discharge. This level of granularity was important for characterizing the multidimensional nature of disability in the early transition phase. The 12-item version does not allow for domain-specific scoring, which would have limited our ability to describe these patterns.

For the follow-up assessment, we opted for the 12-item version to reduce respondent burden and maximize response rates, as patients were no longer in a structured clinical setting. Since both versions measure the same underlying construct and can be converted to a standardized index score, we considered this approach acceptable for comparing overall disability levels over time.

That said, we acknowledge the methodological limitation this introduces. If the study were to be repeated, we would consider using the 12-item version at both time points to enhance comparability and consistency.

Comment: Research tools in sections 2.3.2 and 2.3.3 are not clearly described.

Response: These sections have been revised and expanded to provide clearer and more analytical descriptions of all tools used.

Comment: Practical implications should be further described.

Response: A dedicated section (4.2) now outlines practical implications, including fatigue screening, gender-specific strategies, and person-centered rehabilitation.

Reviewer 3 Report

Comments and Suggestions for Authors

Summary of the manuscript

The authors conducted a prospective observational follow-up study including 137 patients discharged from specialised rehabilitation clinics with Acquired Brain Injury. A baseline assessment was performed, collecting data from an interviewer-based survey one week before discharge, and follow-up was registered with a self-reported survey three months after discharge. Patient-reported outcomes, including disability and health-related quality of life, were collected at discharge and three months later. Finally, they evaluated associations between disability and known risk factors using multiple linear regression.

Overall impression of the work

The manuscript deserves some compliments for its methodology in collecting data and the objective of building a follow-up investigation. The tools the authors planned for the project align with OMS and ICF, so the measures and methods are directly connected with international measures of people’s quality of life.

The results are presented with full details and tables. The authors reported data for all patients, including respondents and non-respondents lost at the follow-up. Interestingly, more information than expected can be obtained from non-respondent patients. Some considerations on the data could still be drawn from the subgroup of patients affected by stroke.  Different-titled paragraphs help the reader in reaching the main information section. A comparison with other studies is reported. They highlighted the importance of paying attention to patients' quality of life and the meaningful life after ABI. The manuscript reported the Danish health-care perspective but could represent an essential guide for planning investigations in other countries and rehabilitation centres. Suitable for the topic of the journal. I hope my suggestions and comments can enrich the paper.

General comments and specific comments:

  1. From lines 44 to 57, the authors interestingly reported the usual path for Danish patients, explicitly reporting “Danish health-care perspective” to other Danish studies. This could be similar to most Northern European rehabilitation but different from other countries. I suggest inserting a “naive” sentence in the conclusion sections, considering if a similar follow-up investigation could be replicated, suggesting further investigation in this direction to give an idea of how important the care assistance is (could be), the care assistance for ABI patients
  2. Lines 82 to 85. The authors defined the period and the place of the recruitment. Challenging period, I am sorry for the strength they need to push into the project (both the authors and the patients). Two considerations arose from those lines: a) I suggest to specify if you have considered that the fear/instability of the pandemic could create some confounds in the HADS on other questionnaries; ii) I suggest to specify if the three clinics has followed common rehabilitation programmes (for sure the authors data collection was precise and coordinate, but also the different rehabilitation discharge could create some differences in patients perception and self perception.
  3. Line 85: “The HNCURC is located in three cities: locations I-III, and eight clinics? I suggest that the authors provide more explanation about the integrated network of clinics and describe how the system of care, assessment, and rehabilitation (including rehabilitative strategies/methods) is integrated. (What an inspiring integration!). Moreover, I suggest checking for the acronyms to be better founded because the organisation seem to be HNRC – maybe I followed the wrong web outcome, but I more than double checked.
  4. Lines 95 and 96: “nor were minor cognitive deficits like attention deficits or mild aphasia.” And line 390-391 “…there was a statistically significant difference in their cognitive function”. I suggest specifying the assessment with which they evaluated those cognitive aspects.
  5. Lines 96-97. I agree and disagree at the same time. Evidence of recovery after TBI shows that microglia are rapidly activated and have pro-inflammatory and neuroprotective roles. The mechanism of accelerated plaque deposition, pathology, and cognitive decline could appear similar to that of a neurodegenerative condition, but the neuroinflammatory processes are different. So I agree with the first part of the sentence, that the symptoms are common, but I would not be so sure that they would likely similarly influence all outcomes. Unless the authors have relevant reference notes, I suggest removing the last part of the sentence.
  6. Line 138. The authors considered the ABI type as a covariate. Could they also have checked in some way for the lesion side, as they reported in Table A1? There are 46 left-sided and 47 right-sided patients. In the assumptions of comparing the left and right groups, did the authors conduct any statistics to check whether they reported similar or different self-reported outcomes? I suggest adding information in Table 2 or Appendix A about the % of left vs right vs both patients of the 62 respondents and the 44 lost at follow-up. The results could be surprising and fascinating to discuss, as the emotional processes among patients are different (or maybe not in  this follow-up study)
  7. Line 169-171: A more detailed explanation of how the authors addressed these aspects would be helpful for readers.
  8. Lines 291 to 297 from the results. Could the authors discuss these results in the discussion section? They mentioned some opinions in the limitation section; have they found any relation with HADS or the lesion side?
  9. Line 400-403: “. For people with ABI, anosognosia or affected self assessment of competence may lead to under-reporting of symptoms, especially in early stages[63].”? - Given this statement, it is natural to ask: Did these patients undergo an assessment of their awareness during their rehabilitation programs? In the evaluations, I noticed no specific measure for anosognosia for hemiplegia was included. Indeed, it would have been challenging to reassess awareness at follow-up, especially considering that the other assessments are self-reported. However, I would have included it as a baseline control, as was done with cognitive tests (to be mentioned). This data would likely have benefited the research, as patients with impaired awareness might report higher scores in the initial phase. In contrast, at follow-up—should awareness improve—they might report lower scores, introducing a potentially disadvantageous bias. The authors mentioned this consideration in the limitations section, but I suggest expanding this section by arguing how important awareness is to assess in patients. Recently, some suitable clinical batteries have been published, in which anticipatory awareness and awareness of activities of daily living are also measured.

Minors:

  1. Line 37: I suggest citing Kılınç S and collaborators (instead of "et al) and inserting the year in brackets
  2. Table 2: I suggest defining a common stylistic choice to describe the Yes/No answer because the range reported for “Pre-morbid characteristics” could be the same for the “Diagnosis, severity and duration”, where the “-“represents the range.
  3. Table 3: I suggest clarifying some terms in the caption (i.e. Yes/No, CI).
  4. Table 2 and Table 4: I suggest evaluating whether it would be helpful for readers to read the % in the c items. The list of numbers in the format number(number) in the column is sometimes hard to follow.
  5. Line 269: where the authors described the disability level in figure 2, I suggest considering it to be consistent with the self-report from the WHODAS 2.0 writing “self-reported level of disability in their daily life” or similar. Moreover, on line 308 of the discussion, the authors write that the patients returned to work “had lower disability levels than non-working patients”. Did they have or did they declare? There is a very subtle but fundamental difference in emotional and self-perception factors, as the authors reported in lines 317-318, “as the patients might take on more complex and challenging tasks at home than in hospital settings”. I am wondering if the authors agree
  6. Please note that references 64, 65, and 66 are included in the bibliography but do not seem to be cited in the main text.

Author Response

General Response to All Reviewers

We sincerely thank all three reviewers for their detailed and constructive feedback. Your comments were not only insightful but also phrased in a collegial and encouraging tone. We found your suggestions to be highly relevant and well-informed, and they provided a solid foundation for improving the manuscript. We have carefully addressed each point and revised the manuscript accordingly. We hope the changes reflect your valuable input and enhance the overall quality and clarity of the study.

Response to Reviewer 3

Comment: Consider adding a sentence in the conclusion about replicating this study in other healthcare systems.

Response: A sentence has been added to the Conclusion, encouraging replication in other international contexts.

Comment: Clarify whether the pandemic influenced responses and whether rehabilitation programs were standardized.

Response: The Methods and Limitations sections now address these points. All clinics followed a standardized model, and potential pandemic-related confounding is acknowledged.

Comment: Clarify the structure of the HNRC network.

Response: Section 2 now includes a detailed description of the HNRC’s integrated structure and rehabilitation model.

Comment: Specify how cognitive function was assessed.

Response: Cognitive function was assessed using the FIM cognitive subscale, now clearly stated in the Methods.

Comment: Revise speculative comparison between ABI and neurodegenerative conditions.

Response: The sentence has been revised to reflect that while symptoms may overlap, the underlying mechanisms differ.

Comment: Consider analyzing lesion side differences.

Response: Thank you for this relevant suggestion that reflects great knowledge on the field. We agree that lesion side can be an important factor in stroke-specific studies, particularly when exploring emotional and cognitive outcomes. However, in the present study, we included a broader ABI population, encompassing not only stroke but also traumatic brain injuries, subarachnoid hemorrhages, and cerebral infections. In these cases, lesion side is often diffuse, multifocal, or difficult to determine due to complex injury dynamics. Therefore, while lesion side was recorded for patients with stroke, we were unfortunately not able to conduct a meaningful subgroup analysis based on lesion laterality across the full cohort. We acknowledge that such an analysis could yield valuable insights in a more homogeneous stroke population, and we will hope for and appreciate your understanding that this was beyond the scope of the current study.

Comment: Clarify covariate analysis and discuss results.

Response: Covariates are presented in Figure 3 and Appendix D. The discussed was updated in general and we hope you find it discussed in relevant detail now.

Comment: Expand discussion on awareness and anosognosia.

Response: The Limitations section now includes a comprehensive discussion of awareness, ethical considerations, and the absence of validated Danish tools. Proxy assessments are suggested for future research.

Minor Comments:

- Citation style corrected (e.g., Kılınç et al.).

- Table formatting and caption clarity improved.

- “Self-reported” phrasing added for consistency.

- References 64–66 are now cited or removed.

Round 2

Reviewer 2 Report

Comments and Suggestions for Authors

Dear Authors,

The revised manuscript has improved substantially. You have responded well to my previous comments and made revisions that enhance both readability and clarity. The study presents meaningful data on disability in patients following acquired brain injury and offers relevant clinical implications. I suggest that the manuscript be accepted for publication after minor changes.

In particular:

  • The introduction is now more straightforward, with better flow and updated references.
  • Definitions of key terms, such as ABI and disability, are more precise and grounded in established frameworks.
  • The aims and methodology are easier to follow after revisions.
  • The use and explanation of different WHODAS versions is now well justified.
  • Practical implications are discussed more thoroughly.

However, there are still some points that should be addressed:

  • While the authors mention the concept of 'Meaningful Life' as central to post-ABI rehabilitation, it remains underdeveloped in the manuscript. There's no clear definition or framework for what constitutes a "meaningful life" in this context. A brief expansion would strengthen the manuscript's understanding.
  • Figure 2 shows individual changes in disability and HRQoL over time. The legend could clarify whether the plots use raw data (direct patient scores) or adjusted data (e.g., controlled for covariates), to assist the reader in interpreting the results.

Author Response

Response to Reviewers and Summary of Manuscript Changes

Response to Reviewer 2

Comment 1: Expansion on the concept of 'Meaningful Life'

Response: Thank you for highlighting the importance of further elaborating on the concept of a “meaningful life.” We agree that this concept is central to the study and to understanding rehabilitation outcomes post-ABI in the described clinical context. In response, we have expanded the introduction and discussion sections to include a clearer framing of “meaningful life,” drawing on the work of Kılınç et al. (2022), the ICF framework and a scoping review from 2013 by Ådal and collaborators.

Change in manuscript:

- Expanded the paragraph in the Introduction (Section 1) to define “meaningful life” in the context of ABI rehabilitation.

- Added a sentence in the Discussion (Section 4.2) to reinforce how the findings relate to this concept.

Comment 2: Clarification of Figure 2 data (raw vs. adjusted)

Response: Thank you for pointing this out. We have clarified in the figure legend that the data presented in Figure 2 are unadjusted, representing raw patient-reported scores at discharge and follow-up. We hope this clarification helps readers interpret the individual-level changes more accurately.

Change in manuscript:

- Revised the legend of Figure 2 to specify that the plots show unadjusted (raw) data.

Reviewer 3 Report

Comments and Suggestions for Authors

I appreciated that the authors addressed the required changes and provided appropriate responses following the first revision action. Overall, I find that they remained consistent with their research approach, responding firmly to the reviewers’ requests and clarifying the boundaries of their study, rather than forcing interpretations that would not align with the scope of their work. I consider these responses acceptable and coherent.

However, it is necessary to refocus attention on the factor of awareness, which I had already raised as point 9 in my previous review action. The authors have correctly understood the concern, but they should be more precise in how they articulate their position.

Lines 482–484: While reporting the need to use a Danish assessment tool is appropriate, the evaluation of AHP can also be performed clinically. It is not inherently inappropriate for patients, as clinicians can conduct the assessment carefully and with targeted questions. For this reason, I recommend revising lines 485–489: the current wording may misleadingly suggest that awareness is inappropriate to investigate in the early stages. However, interview-based assessments can support clinicians and patients in better understanding the situation, while fully respecting the patient's readiness, emotional safety, autonomy, rehabilitation, and home-return safety.

This sentence should be modified. To address potential ethical concerns, the authors could cite international assessments, such as MUNA (doi: 10.1080/13803395.2021.1876842), which explores awareness through general information regarding the patient’s medical history and reasons for hospitalisation (see also Berti et al., 1996; Bisiach et al., 1986). In subacute stages, these assessments can be complemented by evaluations of motor abilities and their impact on daily living activities.

I would like to reiterate that I greatly appreciated this follow-up study, which I find well-structured and clearly described. I insist on this specific point because it is crucial regarding the complexity of the discharge transition phase for patients with ABI, as well as to the study’s findings regarding the perception of quality of life.

I would recommend, in general, reviewing the newly added sections for minor typos. I include specific suggestions in bold below:

Lines 122-123: Please consider to review the sentence “Patients with mild cognitive deficits included, as these symptoms are common after ABI” with this suggestion: Patients with mild cognitive deficits were included, as these symptoms are frequently common after ABI.

Author Response

Response to Reviewers and Summary of Manuscript Changes

Response to Reviewer 3

Comment 1: Clarification on awareness and ethical considerations (Lines 482–489)

Response: We appreciate your thoughtful feedback on the importance of assessing awareness post-ABI. We have revised the relevant paragraph to clarify that while we did not include a formal awareness assessment tool due to the lack of a validated Danish version and ethical concerns, we acknowledge that clinical assessments of awareness can be conducted sensitively. We now reference the MUNA tool (doi: 10.1080/13803395.2021.1876842) and related literature (Berti et al., 1996; Bisiach et al., 1986) to support this point. The revised text emphasizes that awareness can and should be explored in a patient-centered manner, even in early stages, when appropriate.

Change in manuscript:

- Revised lines 485–489 to reflect a more balanced view on awareness assessment.

- Added citations to MUNA and foundational awareness literature.

Comment 2: Minor language correction (Lines 122–123)

Response: Thank you for catching this. We have revised the sentence for clarity and grammatical correctness as suggested.

Change in manuscript:

- Revised line 122–123 to: “Patients with mild cognitive impairment were included, as these symptoms are frequently common after ABI.”

Summary of Manuscript Changes

Section

Change

Introduction (Section 1)

Expanded definition and framework for “meaningful life”

Discussion (Section 4.2)

Linked findings to the concept of meaningful life

Figure 2 Legend

Clarified that data are unadjusted/raw

Lines 485–489

Revised to reflect a more nuanced view on awareness assessment; added references to MUNA and others

Line 122–123

Corrected sentence on inclusion of patients with mild cognitive deficits